# The association of diminished quality of life of Afghan adults' psychosocial wellbeing, in the era of the Taliban 2.0 government

Jessi Hanson-DeFusco[1]*, Anton Sobolov[2], Sami Stanekzai[3], Alexis McMaster[4], Hamid Popalzai[5], Heer Shah[6], Min Shi[7], Nandita Kumar[7]

1 Humanities, Social Sciences and Communication Department, Lawrence Technological University, Southfield, Michigan, United States of America, 2 School of Economic, Political and Policy Sciences (EPPS), University of Texas at Dallas, Richardson, Texas, United States of America, 3 EPPS, University of Texas at Dallas, Richardson, Texas, United States of America, 4 Graduate School of Public and International Affairs, University of Pittsburgh, Pittsburgh, Pennsylvania, United States of America, 5 Afghanistan Technical Vocational Institute (ATVI), AIT Compound-MoHE, Kart-e Char, Kabul, Afghanistan, 6 Department of Biological Sciences, University of Texas at Dallas, Richardson, Texas, United States of America, 7 EPPS, University of Texas at Dallas, Richardson, Texas, United States of America

* jhansonde@ltu.edu

**Data Availability Statement:** There are ethical restrictions on sharing a de-identified data set, as

## Abstract

After the 2021 US withdrawal, a drastic transition of power coupled with international sanctions and the Islamic State-Taliban conflict led to growing issues of widespread economic hardships, food insecurity, stricter social policies, and changes to daily life. This 2023 study examines the association of diminished quality of life (DQOL) on the psychosocial wellbeing of Afghan adults living in-country under the Taliban- Islamic Emirate of Afghanistan. Applying Solar & Irwin's social determinants of health framework, we present the quantitative analysis of data collected from 873 Afghan respondents (ages 18–65) of a digital survey, using snowball sampling over social media. Data analysis examines the association between individual self-reported quality of life hardships and psychosocial stress symptoms (disaggregated and aggregated), disaggregated by demographics. Approximately nine-in-ten Afghans face DQOL correlates related to higher psychosocial stress (PSS). 72.9% (CI95% 69.8–76.0) of respondents self-report suffering food insecurity; 71.6% (CI 95% 68.3–74.8) poor access to needed healthcare. The extent to which Afghan men face limited household healthcare access is linked to higher PSS levels ($\chi2 = 117.10$, $p<0.001$). A matching analysis of survey data indicates that Marginal Effects that lack of healthcare access increases the probability of stress by approximately 8%; experiencing the loss of loved ones also has a significant effect ranging from 9% to 11%; and experiencing threats of violence leads to a substantial increase in the probability of PSS, ranging from 34% to 36%. Qualitative data triangulate the statistical findings, provides intrinsic insight into Afghans' daily experiences, and inform causal mechanisms related to share trauma experiences. The 2021 US withdrawal marked a turbulent political shift in Afghanistan that disrupted previous structural determinants of health, like gender and age. The political shift, international sanctions, and internal crises have worsened the humanitarian conditions affecting most Afghans, negatively impacting their physical and psychosocial wellbeing.

data contain potentially identifying or sensitive participant information, as mandated by the University of Texas Institutional Review Board and Office of Human Subjects Protections (OHSP) (contact information- research@utdallas.edu). Additionally, while the study was anonymous, the participants of this study were informed that their data would not be shared publicly, thus they did not give written consent. Due to the sensitive nature of the research, supporting data is not available.

**Funding:** NK and HS received the 2023 Hobson Wildenthal Honors College Undergraduate Research Apprenticeship Program (URAP) grant to serve as co-researchers on this project, from the University of Texas at Dallas (https://honors.utdallas.edu/research/undergraduate-research-apprenticeship-program/). The funders had no role in study design, data collection and analysis, decision to publish, or preparation of the manuscript. No other authors received specific funding for this work.

**Competing interests:** The authors have declared that no competing interests exist.

## Introduction

On August 30, 2021, the United States (US) finally withdrew its military forces from Afghanistan after nearly 20 years of the US- North Atlantic Treaty Organization (NATO) involvement in the 'War on Terror,' which the Taliban historically has referred to as an occupation. The withdrawal was followed by a swift collapse of the Islamic Republic of Afghanistan, under President Ashraf Ghani. As a Taliban offensive moved on Kabul, tens of thousands of Afghans desperately tried to flee [1–4]. 8.2 million Afghans are refugees, including 1.6 million who have fled since 2021. Yet 28.3 million continue to reside in their homeland, with over 9 million internally displaced due to conflict. Nearly two-thirds of Afghans are believed to be affected by the growing humanitarian crisis [5]. Hibatullah Akhundzada became the supreme commander of the Taliban, in May 2016. After the 2021 takeover, he took the role as prime minister of the nation, which the Taliban called the Islamic Emirate of Afghanistan. The cabinet was led by Mullah Mohammad Hassan Akhund [6]. There are different opinions on whether this new government (often termed New Taliban/Taliban 2.0) will be as radical in its ideologies and governance as the Taliban regime of the 1990s, or whether it may be different. In any case, the rise of the New Taliban regime has led to drastic changes in governance, political and socio-economic policies, and daily life for Afghan citizens. Additionally, international sanctions on the regime, inter-ethnic conflict, along with the Islamic State (ISS)-Taliban conflict within country exacerbate growing economic hardships, along with deterioration of the nation's infrastructure and public service provision [1,3,4,7–9]. Yet, it is difficult to know to what extent.

In the months following the US withdrawal, the (so-called) Islamic Emirate of Afghanistan under the Taliban initially allowed international nonprofit organizations (INGOs) to remain to help in the transition. Most foreign stakeholders pulled out, but some remained like Doctors without Borders and the Red Cross (IRCR), which permitted information sharing from within the country. Studies that were conducted before 2022 indicate a devastating decline in healthcare and public services quality, including less access to medicines, functioning clinics and hospitals, medical staff, and critical health services contracted between Afghanistan's Ministry of Public Health (MOPH) and nonprofit organizations (NGOs) [10–14]. Just after the withdrawal, approximately 80% of surveyed Afghans reported experiencing some type of change in their lifestyle after the Taliban regained control [12].

Initially there was debate about whether life under this government might be any better. While international sanctions were put in place, some regional countries like Russia, India, and China maintained political and economic activity with the Taliban 2.0 but refrain officially acknowledging its legitimacy [15]. In early 2022, many Afghans said "the country fe[lt] safer, less violent than it has in decades, but the once aid-fueled economy [was] barreling toward collapse" [16]. Many Afghans seemed to accept the Taliban 2.0 rule, often preferring peace and stability even at the sacrifice of democratic freedoms [7,9,17]. However, the socio-economic conditions for men, women, and children in general continued to worsen [1,18]. Unemployment grew. Additionally, the Islamic Emirate of Afghanistan government implemented more stringent policies on most of the population. There were growing reports of reprisals by the Taliban mainly towards male citizens who had supported US stakeholders [14,18,19]. In early 2022, food insecurity rose as the West instigated more sanctions and a harsh winter led to more shortages. Many households faced severe malnutrition and starvation [15,20,21]. Health research indicates that all these challenges can readily diminish Afghans' quality of life (QOL) [3,5,7,8,22], acting as social determinants of health (SDH) and mental health (SDMH).

In global health, SDH frameworks build on the premise of *social gradient*, which indicates that people with diminished socio-economic status face increased health risks and lower life

expectancy. Similarly, negative conditions can lower QOL resulting in poorer health outcomes [23–25]. "Considering mental health, the social gradient impacts both risk of disorder and access to services" [25]. Experiencing hardships, including malnutrition, poverty, conflict, and violence, is frequently associated with higher levels of psychosocial stress [26–29]. Afghan citizens have long-faced issues of poor mental health, including generational trauma. Due to the nation's various conflicts (1979–2023), a large majority of Afghans likely suffer from one or more forms of mental disorders [10,13]. The 2021 power shift and ongoing humanitarian crisis likely compounded this trauma.

Under the first Taliban rule (1996–2001), approximately 97% of women were estimated to suffer depression, 86% anxiety, and 42% met the diagnostic criteria of post-traumatic stress disorder (PTSD) [30,31]. In the years following the American military interventionI in 2001, studies estimated ongoing high rates of depression (36.5–67.7%), anxiety (51.8–72.2%), and PTSD (20.4% to 42.1%) among all Afghan adults, but with odds ratios higher for women compared to men [30,32]. Due to diminished social status related to gender inequality and misogynistic policies of the regime, Afghan women tended to have a higher probability of mental illness due to stigmatization and less access to care [8,30,31].

Likewise, numerous studies found that elderly-status also influenced higher stress levels, although not as significant as gender. A 2015 study of Afghan refugees indicates that older age (paired with being a woman) at the bivariate level linked to higher levels of perceived stress were associated with psychological distress [33]. Daily stress increases among older Afghan individuals, often related to age-related physical health complications (i.e. high blood pressure) and additional physical hardships like violence, food insecurity, which many experienced in previous wars [34–36]. A 2021 meta-analysis implies that elderly survivors of war frequently experience higher prevalence rates of cognitive impairment, comorbid disorders including head injuries and depression, and medical comorbidities. These age-related issues correlate to poorer QOL and social hardships [37]. In Fall 2021, nearly 36% of surveyed females reported extreme concern for their physical health compared to 16% of males (p<0.001). 54% of females also expressed having extreme concern for their mental wellbeing compared to 22% of males (p<0.001) [12]. Comparatively, a late 2021 study reports that 80.4% of Afghan women exhibited depression symptoms, while 81.0% present mild to extremely severe anxiety [22]. Such studies provide some reference to mental health/psychosocial stress (MHPSS) but only in the shadow of the US withdrawal.

In late 2022, the Taliban Islamic Emirate of Afghanistan implemented more extremist policies including banning women from working, forcing nearly all remaining foreign entities to cease Afghan operations. With the exit of external actors, it has become increasingly difficult to extract information from within the country as conditions continue to deteriorate. A review of the literature indicates that while there are numerous contemporary studies on the association of QOL and psychosocial health of Afghan refugees, there is considerably less current research concerning Afghan domestic populations. Furthermore, health scholarship indicates that prolonged exposure to diminished quality of life (DQOL) can exacerbate poor MHPSS among affected peoples [11,23,38–40]. Against this backdrop, our study seeks to offer a more contemporary analysis of DQOL experiences and psychosocial stress (PSS), years into the transition of power. We present 2023 digital survey data collected from 873 respondents living through the humanitarian crisis in their homeland. In the next section, we present the methodology, including the theoretical framework, survey design, data collection, and analysis. The rest of this article is presented as follows: the Results section presents the statistical findings (correlation analysis, t-test analysis, multivariate regression analysis, logistic (odds-ratio), and matching modelling) related to the research questions. The Discussion section explores the quantitative findings in more detail related to relevant scholarship as well as qualitative

statement made by respondents- the purpose of which is to triangulate the findings and to help provide a deeper insight to the causal mechanisms [33,41,42]. We conclude the paper by presenting research implications and limitations, along with proposed policy recommendations.

## Material and methods

### Ethics statement

This study received ethical approval from the University of Texas-Dallas IRB (ID#22–648) before data collection. Formal consent was obtained by participants through an electronic informed consent process. The IRB informed consent information was provided at the beginning of the survey in all three languages (English, Pashto, Dari). Participants voluntarily agreed to consent to participating in the study by clicking on the prompt "Yes, I consent to participate" or could click "No, I do not consent." No identifying information including names was collected to safeguard participant anonymity.

### Purpose of this study

This study examines the following questions:

1. To what extent do Afghan adults (ages 18–65), living in the era of the Taliban 2.0, self-report experiencing ongoing DQOL (measured by disaggregated common hardships and aggregated levels)?

2. To what extent does DQOL (aggregated and disaggregates experiences of poorer access to: a) food,b) healthcare, c) infrequent contact with loved ones, d) lost family, and e) threat of violence affect their psychosocial stress (measured by the Perceived Risk Scale; aggregated and disaggregated by gender, age range, household conditions, and religiosity)?

Based on the commonalities within relevant literature, we test the following hypotheses:

H1: Afghan adults experiencing higher rates of physical hardships related to behavioral & health system factors (food insecurity, violence, & limited healthcare access) are correlated to higher PSS (individual stressors and aggregate)

H2: Afghan adults who experience high rates of psychosocial circumstances (loss of loved ones, limited social networks) are correlated to higher PSS

H3: Those who experience higher rates/frequency of DQOL factors and aggregated DQOL exhibit higher rates of PSS

### Theoretical framework

This study theoretically draws from the World Health Organization's Commission on Social Determinants of Health (CSDH) Framework, developed by Solar & Irwin (2010). "Effects of social determinants on population health and on health inequalities are characterized by working through long causal chains of mediating factors [that can] cluster among individuals living in underprivileged conditions" [24]. The CSDH Framework identifies two types of social determinants of health/mental health and how they interact (see Fig 1). Firstly, *Structural Determinants* include socio-economic and political contexts in which social hierarchies exist, often maintained through governmental policies, formal institutions, economic structures, and education systems. Thus, structural mechanisms are set, such as income, gender, and education level. These mechanisms, the context in which they operate, and a person's socio-

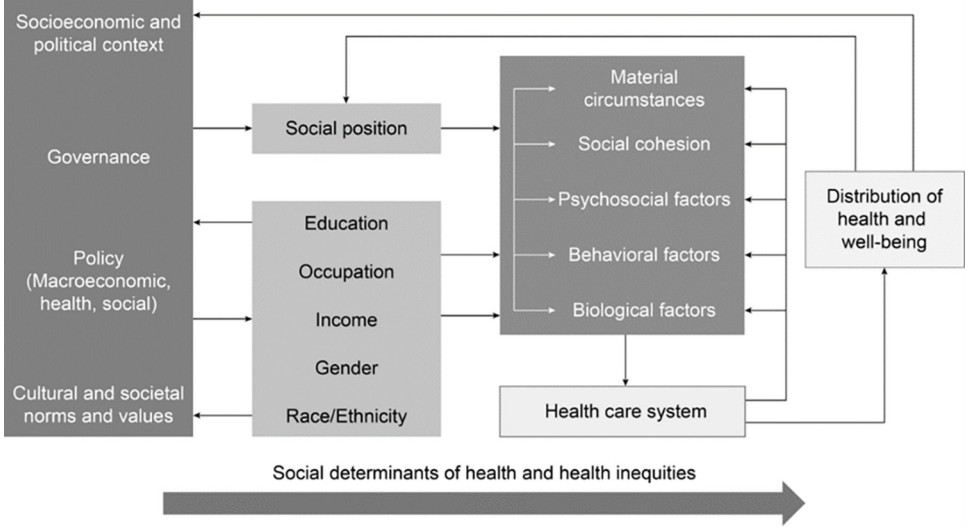

**Fig 1.**

economic circumstance that comes as a result, make up social determinants of health. These primary determinants further inform the differences in individuals' risk and exposure to a second type- *Intermediary Determinants of Health (IDH)*. These secondary determinants influence the kinds of vulnerability and exposure to negative factors that directly impact the health outcome of people. Intermediary determinants include behavioral factors (e.g. nutrition, physical activity), health system factors (access and quality of services), psychosocial circumstances (e.g. lack of social support networks, loss of loved ones), as well as material circumstances (e.g. housing, physical working space) and biological factors (e.g. genetics, disabilities).

## Context

This study used a mixed-method design, utilizing a semi-structured snowball survey (on the Qualtrics platform) of Afghan adults residing in country, launched over social media. The project, entitled *Diminished quality of life among adult civilians affected by the 2021–23 Afghanistan humanitarian crisis*, is one of the only known psychosocial stress survey studies conducted in Afghanistan since the Taliban took full control over the national government in 2022–23. Our research primarily concerns assessing IDH behavioral factors, health system, and psychosocial circumstances. The conditions in the field make it difficult to establish baseline data before the US withdrawal; thus, we look for key trends found in relevant literature preceding the timeframe of our study. We base our three initial hypotheses on these trends. The primary investigator and co-investigators designed the survey using previous studies on psychosocial stress in conflicts and disasters, including using a modified perceived stress scale (PSS). The purpose of the data is to capture changes in the living conditions not only due to the change in government, but moreover, the impending humanitarian crisis exacerbating the economic hardships on Afghans (pro-Taliban and pro-US civilians alike) due to the sanctions imposed by Minority World nations (with less than 20% poverty rates) like the US.

## Data collection process and study population

A robust literature review of 231 top-cited studies on Afghan QoL was performed on Google Scholar and PubMed (using search terms including *Afghanistan*, *humanitarian crisis*, *war*,

*mental health*, *psychosocial wellbeing*, *PSS*, *quality of life*, *study)* by the PI and CIs to identify key QOL variables for the analysis, as well as an in-depth contextual understanding of the political, economic, and social history of Afghanistan. The survey closed-ended and open-ended questions were translated and piloted with lead facilitation by our Afghan co-authors before launching data collection, including assessing for contextual and cultural relevance. Approximately a dozen adult Afghans living in the host-country and abroad voluntarily participated in the pilot-test, with all feedback incorporated to minimize threats to validity, including feedback to limit any demographic questions pertaining to ethnicity or direct political affiliations as they were often perceived as triggering. Pilot tested data was excluded from the analysis.

The survey was translated and presented in Pashto, Dari, and English (translation verified by Afghan translator and pilot-testing process). As part of the ethical approval process, we selected to present the study design and survey electronically with the Ministry of Higher Education and the Ministry of Public Health (MoPH) of the 2022 Afghan Government to ensure adherence to ethical protocols to responsible practices and safeguard of participants' privacy and safety given the political climate and strict media regulations. No formal response was received by the Afghan ministries. Data analysis indicates that numerous participants who took the survey worked for the Afghan government or military and/or held pro-Islamic Emirate of Afghanistan affiliations. Participation eligibility criteria included fluency in at least one language, being between the ages of 18–65, residing within Afghanistan at the time, and completing the IRB consent process. Data collection occurred from July to December of 2023, using a snowball sampling design of sharing the recruitment script in all three languages over Facebook (in 2021 approximately 68% of Afghans own mobile phones/iPhones and millions use social media, with subscription increases in 2022 [43,44]). The survey was then shared on over 50 publicly open Facebook pages related to Afghan news, regional/community pages, university sites, education boards, governmental sites, and human rights pages. Participants could move between questions, and decline to answer all open-ended questions, and select 'decline to answer' or 'n/a'.

## Instrumentalization

We designed and launched a digital survey with both close-ended/multiple choice questions related to PSS, demographics, and DQOL factors, and additional open-ended questions where participants could elaborate if they chose, or they could leave it blank. No interviewers were used, and the open-ended questions were based on similar DQOL questions from our research, as well as adapted from USAID-DHS survey applied in similar Afghan studies [45]. The digital survey includes multiple choice questions measuring participants' self-reported demographics (age, gender, resident within country status, household size), QOL factors, and PSS levels (translated in all three languages) (see S1 Text for all questions). Survey questions related to MHPSS were based on Foa (2001) psychosocial stress scale (PSS), often used for participant self-reporting stress, [46–48]. This instrument was also adopted, modified, and tested in various studies including in post-war, crisis in low-income nations including Afghanistan, Ukraine, and Liberia [46–52]. PSS scales are regularly used to assess psychosocial stress self-reported by participants experiencing increased trauma events [26,28,46,47]. These scales capture individual stressors like anxiety, extreme anger, or withdrawal, using Likert scaling, which can be assessed both disaggregated and aggregated, testing Cronbach's alpha for inter-reliability. Additionally, QOL factors are also assessed using a Likert scale, assessing both individual and aggregated levels (see Table 1).

**Table 1. The reliability test of study variables & scales.**

| Variable | Description | Cronbach's α | Pearson R coefficient-overall score |
|---|---|---|---|
| **PSS Level** | **Total Likert score of all stressors (0 none to 21 high)** | **0.89** | |
| Sadness | Person's weekly experiences (0 never, 1 some days, 2 most days/week) | 0.88 | 0.77*** |
| Anxiety/worry | | 0.86 | 0.82*** |
| Withdrawal/ avoiding people | | 0.87 | 0.77*** |
| Anger | | 0.87 | 0.78*** |
| Poor sleeping habits | | 0.86 | 0.81*** |
| Poor eating habits (too much or too little) | | 0.88 | 0.74*** |
| Nightmares | | 0.88 | 0.74*** |
| **QoL hardship level** | **Aggregate total of factors (0 none-10 high)** | **0.70** | |
| Not able to buy enough food | (0 never, 1 some days, 2 most days/week) | 0.65 | 0.69*** |
| Not able to access healthcare services or medicine | | 0.64 | 0.70*** |
| Infrequent contact with friends and family outside of HH | | 0.66 | 0.67*** |
| Face threat of harm or violence | | 0.65 | 0.71*** |
| Bereavement for lost loved ones | | 0.67 | 0.62*** |

Note: ***$p < 0.001$. Qol = quality of life factor. PSS = psychosocial stress symptom.

## Participants

Of the 1798 people who opened the survey, 891 respondents consented to take part and provided self-reported information. 22 were excluded as they stated that they resided outside of Afghanistan, along with 26 respondents who did not disclose their location. This study only considers within-country populations at the time of the study. An initial review of open-ended question responses indicates a balanced distribution of participants who worked for or held political affinities for the deposed government and the American forces, compared to those who work for or are in favor of the Taliban 2.0 government/ Islamic Emirate of Afghanistan. Data on specific location, language, and ethnicity were not collected for ethical reasons presented by Afghan consultants, who urged that such questions would make many respondents feel uncomfortable disclosing and/or fearful of the study's intent in current political climate and the historical conflict between specific ethnic groups. Future studies may consider including additional demographic variables. This analysis purposefully examines Afghans as a country of citizens under a new era, often facing similar hardships irrespective of their political or ethnic background.

## Data analysis

For this mixed-method study, we conducted a statistical analysis using Stata, including descriptive statistics, pairwise correlation analysis, and multivariate regression modelling of all quantitative survey questions. Based on the snowball method applied, the survey is not randomized, limiting data generalizability. Thus, we further adjust for covariate imbalances through OLS and Matching to further test casual patterns asserted in the unadjusted analysis. The integrity of any observational study hinges on the premise that the treatment and control groups are comparable on all observed covariates [53–55]. Recognizing some initial imbalances in our dataset, we embarked on a two-pronged approach to mitigate these discrepancies.

First, we adjusted for covariates within our regression models. This process is pivotal for controlling confounding variables that could otherwise distort the true effect of the treatment on the outcome. The adjusted model showed improvements in covariate balance, indicating a reduction in potential biases that could affect the results. Second, we employed a matching strategy to create a subsample of treated and control observations that were more alike, mirroring the covariate distribution in the treatment group across the control group, hence reducing selection bias. Matching involves pairing individuals from the treatment group with those in the control group based on similarity across a set of covariates, thereby reducing baseline differences.

Thirdly, we transitioned from traditional regression coefficients to Marginal Effects, focusing on the Average Treatment Effects (ATE) and their Standard Errors (SE). This transition allows us to convey the impact of the independent variables on the probability of experiencing increased PSS in a more intuitive manner. Marginal Effects were calculated using the 'mfx' package in R, which simplifies the process of converting logistic regression coefficients into average marginal effects. This method involves estimating how a slight change in each predictor variable, holding all other variables constant, would affect the probability of the outcome variable.

Lastly, we conduct a qualitative analysis of participant responses to open-ended survey questions to triangulate the quantitative findings, and to explore in depth the meaning behind their life experiences, their struggles, and needs living in Afghanistan after the withdrawal of American forces [33,41,56,57]. The qualitative data analysis was conducted using Excel, in which for data cleaning, notes taking, capturing emerging trends, including coding and tracking these themes. The research team assessed in weekly sessions to analyze and explore these themes together. All data analysis, discussion, and conclusions were supported by all co-investigators, with our Afghan colleagues' interpretations, translations, and expertise prioritized. The qualitative findings are elaborated upon in the Discussion.

## Researcher characteristics and reflexivity

The research team comprised academic researchers, university student researcher assistants, and nonprofit practitioners from the United States, Afghanistan, Russia, and China. Table 2 describes the diversity and strength of all the authors informing the credibility of the results. All authors have social science academic backgrounds. Two co-authors are Afghan, and led in the translation, validation, pilot testing, and data analysis including the qualitative analysis of the survey including open-ended questions. One researcher is directly impacted by the conditions in Afghanistan, including facing violence, limited work, and food insecurity. The second is a student in the US with family and colleagues still residing in Afghanistan. The US-based academics supported the design in English, data collection and analysis. Most have experience conducting rigorous mixed-method, survey research on MHPSS and DQOL in low-income settings affected by extreme poverty, civil conflict, and humanitarian hardships. All have some academic training and work experience in Western institutions. We created a plan to avoid a neocolonial research dynamic on this project and meet to discuss our progress and any concerns. Our diverse cultural backgrounds help provide a robust lens to this research, but we prioritize the expertise, analysis, and feedback of our Afghan co-authors in this entire research process and analysis.

## Results

### Unadjusted quantitative analysis

Of the 843 survey participants who met all criteria to be included in this analysis, 794 (94.2%) self-report being men, 28 (3.3%) women, and 21 (2.5%) do not disclose. 53.7% are aged 18–29;

**Table 2. Characteristics of team members.**

| | |
|---|---|
| JD | American PhD in policy and global health policy; professor in American university; female; Caucasian; 20 years of international development work experience in Africa; Latin America; Asia in human rights and mixed-method field research; focusing on MPHSS and DQOL needs during civil crises; supported development project support and research on Afghanistan for three years |
| AS | Russian PhD in political science; Caucasian; male; advanced statistical analysis skills; professor in American university researching questions of politics using text analysis; machine learning; and causal inference; research projects focus on mass protest; cybersecurity; and political control in autocracies; moderate familiarity with Afghan politics |
| SS | Afghan graduate student at US university; male; former Afghan policy advisor in Kabul before 2021 US withdrawal; family still in Afghanistan and US; fluent in major Afghan languages; research focus on political science; well-connected on social media with political channels; expertise in working with international institutions in development and policy projects in Afghanistan |
| AM | American nonprofit practitioner; 3 years' experience working in Africa; little experience with Afghan issues; led mixed method field research in low-income countries; master's degree in human rights and nonprofit; white; female; experience in conducting research on MHPSS and DQOL in conflict |
| HP | Afghan; male; BA from Afghan university; former translator for US nonprofits; Pashto; living in Afghanistan; fluent in multiple Afghan languages; family impacted by humanitarian crisis in country; expertise on conducting research in country and online in collaboration with national and international co-researchers |
| HS | Indian American; female; pre-medical university and biology student in USA; focus of research is in healthcare policy and global health equity; volunteering to support nonprofits focused on stopping gender-based violence and humanitarian crises including in Uganda, Rwanda, and Afghanistan |
| MS | Chinese; female; PhD in political science with background in machine learning, deep learning, time-series, and other quantitative statistical models to evaluate the competitive trade relations between the US and China; support in co-authoring research related to DQOL, health crises, and conflict |
| NK | Indian American; female; university student in political sciences; research interests in improving global health inequities in USA and low-income countries in MENA and south-east Asia; research experience on humanitarian crisis in Afghanistan |

34.2% are 30–39; 9.7% are 40–49; and 2.4% are 50–65. The average household size is 11.7 people (95%CI 11.3–12.0). This study's findings imply the extensiveness of Afghans' self-reported diminished QOL approximately two years after the US withdrawal. On average, their DQOL levels appear somewhat high (4.2/10). Overall, 88.38% of participants report suffering some level of food insecurity; 88.78% have limited or no healthcare access; 83.59% have infrequent contact with family and friends; 84.82% experience threat of violence, and 71.97% lost at least one or more family (killed or displaced) since the American withdrawal. A t-test analysis indicates that there are no significant gender-related differences between aggregated and disaggregated QOL experiences or PSS levels (t<1.52, p>0.10). Likewise, people aged 50 or older experience these QOL hardships and stress at about the same level as younger Afghans (t<1.60, p>0.10). The distribution of how frequently Afghans experience these hardships varies (see Graph 1), indicating a pervasiveness related to the recent changes. But to what level might these hardships statistically impact stress levels?

Table 3 provides an overview of aggregated and disaggregated DQOL factors and demographics in relation to the outcome of psychosocial stress levels and disaggregated stressors. The survey allowed participants to skip items. Thus, this analysis considers total PSS levels only for participants who self-report on all 7 stressors (n = 597).

On average, reported PSS levels are moderately high (12.23 of a potential 21), with anxiety, poor sleep, and anger being the most prominent stressors. A correlation analysis further indicates that higher DQOL (aggregate and disaggregated factors) are significantly correlated with higher levels of stress and experiencing higher rates of each individual stressor (p = 0.00). The threat of experiencing violence at a higher frequency is highly correlated with high stress levels (r = 0.61, p = 0.00) (see S1 Table).

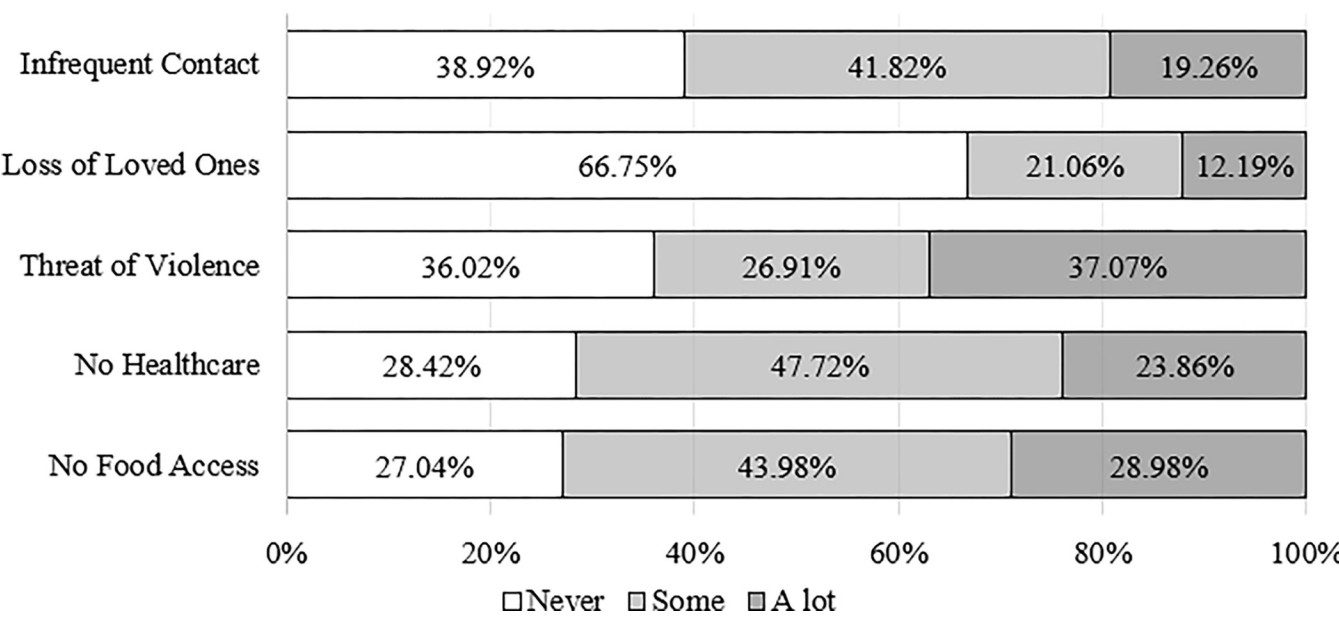

**Graph 1. Diminished quality-of-life factors reported by Afghan adults under Taliban- Islamic Emirate of Afghanistan.**

In this natural study, we further disentangle the individual factors of diminished quality of life and how they shape the channels of the psychosocial stress (aggregate). A glimpse at the raw data indicates a strong relationship between DQOL factors and observed levels of stress among the respondents (see Fig 2 for the discovery analysis).

Multivariate regression modelling indicates that individual PSS levels among Afghan adults who report experiencing higher PSS because of the political shift are significantly related to nearly all DQOL factors, except infrequent contact with friends and family (see Table 4). Age and sex are not significant factors related to stress. In fact, a t-test analysis assessing for

**Table 3. Descriptive statistics of study variables.**

| Variable | N | Range | Mean | SE | 95% CI |
|---|---|---|---|---|---|
| **PSS Level** | 597 | 0–21 | 12.23 | 0.24 | 11.75–12.71 |
| Sadness | 635 | 0–3 | 1.66 | 0.04 | 1.57–1.75 |
| Anxiety/worry | 638 | 0–3 | 2.05 | 0.04 | 1.97–2.15 |
| Withdrawal/ avoiding people | 628 | 0–3 | 1.58 | 0.05 | 1.49–1.67 |
| Anger | 631 | 0–3 | 1.72 | 0.04 | 1.63–1.80 |
| Poor sleeping habits | 625 | 0–3 | 1.89 | 0.04 | 1.81–1.97 |
| Poor eating habits | 623 | 0–3 | 1.75 | 0.04 | 1.67–1.83 |
| Nightmares | 625 | 0–3 | 1.58 | 0.04 | 1.49–1.66 |
| **QoL level** | 747 | 0–10 | 4.22 | 0.09 | 4.04–4.41 |
| Not able to buy enough food | 773 | 0–2 | 1.02 | 0.03 | 0.97–1.07 |
| Not able to access healthcare services or medicine | 767 | 0–2 | 0.95 | 0.02 | 0.90–1.01 |
| Infrequent contact with friends and family outside of HH | 758 | 0–2 | 0.80 | 0.03 | 0.75–0.85 |
| Face threat of harm or violence | 758 | 0–2 | 1.01 | 0.03 | 0.95–1.07 |
| Lost or separated from loved ones | 755 | 0–2 | 0.45 | 0.03 | 0.40–0.50 |
| **Demographics** | | | | | |
| Age (yrs.) | 843 | 18–65 | 30.16 | 0.29 | 29.59–30.73 |
| Household size (total persons living in compound) | 821 | 0–20 | 11.67 | 0.18 | 11.31–12.02 |

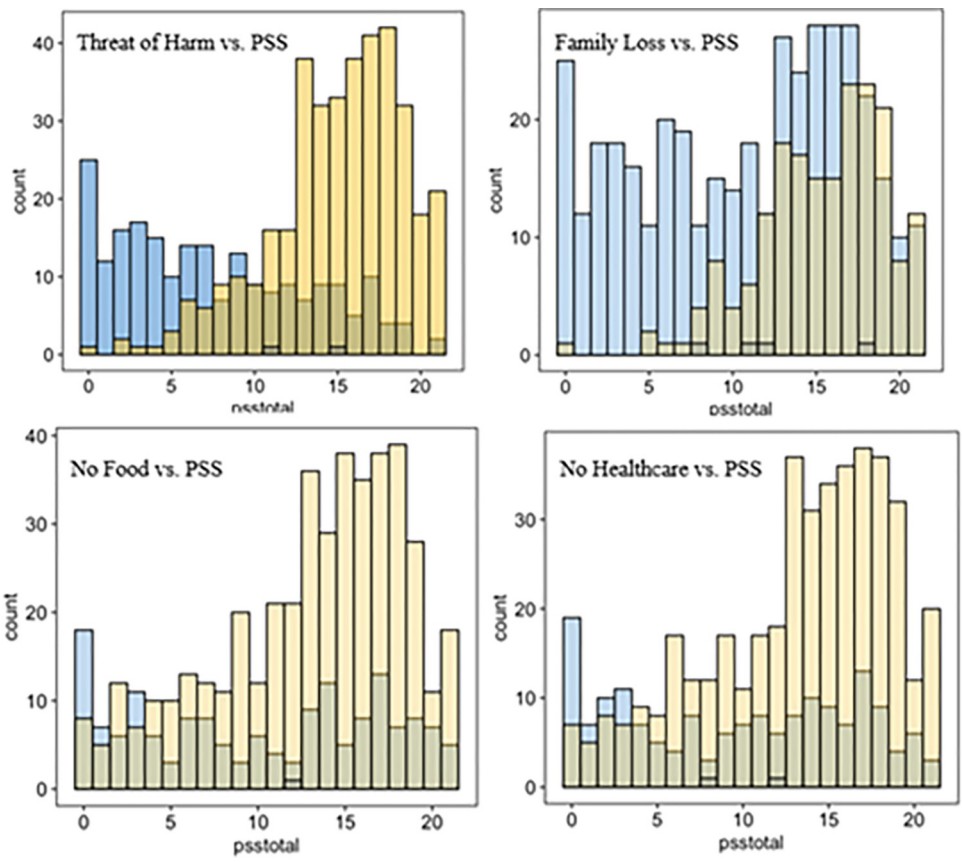

**Fig 2.**

significant differences between men and women finds no significant differences in DQOL factors or PSS (aggregated and disaggregated). The most crucial factor appears to be the extent to which a person faces the ongoing threat of violence or personal harm. Model 1–6 imply that an individual who experiences nearly daily insecurity is likely to exhibit over a third more psychosocial stress than a peer who faces no violence (adj R2≥0.39).

Comparatively, living with an increased number of people in the same household (HH)/compound can slightly lower stress. Models 2 and 3 imply that if a person lives with 20 other direct and extended family members, s/he may experience an approximate 9.5% reduction in stress (adj. R2 = 0.40). Spiritual comfort (mainly from the Muslim faith) also has a significant impact on reducing stress (see Model 4).

Next, we assess if the patterns asserted within the raw data analysis hold up when adjusting for other potential influencers, such as the independent channels (e.g. age, gender), through a battery of empirical tests.

**Sample adjustment analysis.** The matching process results in a sample that is more representative of the general population, as evidenced by an enhanced balance in covariates post-matching. This improvement bolsters the credibility of our findings, ensuring that the observed effects are attributable to the treatment rather than pre-existing differences between groups. A comparative analysis of the unadjusted and adjusted samples highlighted the effectiveness of our methodological interventions. Covariate balance, as measured by standardized mean differences, improved significantly in the adjusted samples, underscoring the importance of these adjustments in reducing biases (see S1 Fig).

**Table 4. Multivariate regression analysis of total PSS of Afghans under Taliban 2.0 & humanitarian crisis by quality-of-life factors.**

| Variables | (1) | (2) | (3) | (4) | (5) | (6) |
|---|---|---|---|---|---|---|
| Can't buy food | 0.61** | 0.63** | 0.61** | 0.51 | 0.52* | 0.70** |
| (never 0–2 a lot) | (0.31) | (0.31) | (0.31) | (0.32) | (0.31) | (0.33) |
| No healthcare | 0.69** | 0.71** | 0.72** | 0.75** | 0.77** | 0.60* |
| (never 0–2 a lot) | (0.33) | (0.32) | (0.32) | (0.33) | (0.33) | (0.34) |
| Threat of violence | 3.69*** | 3.60*** | 3.62*** | 3.62*** | 3.60*** | 3.50*** |
| (never 0–2 a lot) | (0.26) | (0.27) | (0.26) | (0.27) | (0.27) | (0.29) |
| Loss of loved ones | 0.65** | 0.73** | 0.71** | 0.76** | 0.80*** | 0.75** |
| (never 0–2 a lot) | (0.30) | (0.30) | (0.30) | (0.30) | (0.30) | (0.33) |
| Infrequent contact | 0.03 | 0.06 | 0.07 | -0.07 | -0.04 | -0.05 |
| (never 0–2 a lot) | (0.30) | (0.30) | (0.30) | (0.30) | (0.30) | (0.31) |
| HH total | | -0.09*** | -0.10*** | | | |
| (never 0–2 a lot) | | (0.04) | (0.04) | | | |
| Religious comfort | | | | -0.63** | | |
| (never 0–2 a lot) | | | | (0.28) | | |
| Age (years) | | 0.01 | | | | |
| | | (0.02) | | | | |
| Sex (M-1; F-2; | | -0.32 | | | | 0.21 |
| n/a- blank) | | (1.02) | | | | (1.12) |
| Religious comfort* | | | | | -0.04*** | -0.04*** |
| HH (fixed effects) | | | | | (0.02) | (0.02) |
| Age (interaction) | | | | | | X |
| Constant | 6.99*** | 9.30* | 8.11*** | 8.24*** | 8.04*** | 6.65 |
| | (0.43) | (5.12) | (0.58) | (0.67) | (0.54) | (5.98) |
| Observations | 591 | 590 | 590 | 566 | 565 | 565 |
| R-squared | 0.40 | 0.41 | 0.41 | 0.41 | 0.41 | 0.45 |
| Adj. R2 | 0.39 | 0.40 | 0.40 | 0.40 | 0.41 | 0.41 |

Robust standard errors in parentheses. *** p<0.01, ** p<0.05, * p<0.1.

This secondary analysis presents the effects of different factors on PSS, now expressed through Average Treatment Effects (ATE) and their Standard Errors (SE) across four different models: Unadjusted, Adjusted with Controls, Adjusted with Matching, and Combined Adjustment with Controls and Matching. The ATE represents the average change in the probability of self-reporting experiencing PSS across all observations when the predictor variable changes by one unit or switches from 0 to 1 in the case of binary variables. Table 5 summarizes the ATEs and their SEs for each predictor variable across the four models.

Table 5 presents the Marginal Effects, focusing on the Average Treatment Effects (ATE) and their Standard Errors (SE). Firstly, the ATEs for the inability to afford food appear not statistically significant across all models, suggesting that when controlling for other factors, the direct impact of this variable on increased perceived stress is minimal. Secondly, access to healthcare emerges as a significant predictor, with ATEs indicating that lack of healthcare access increases the probability of stress by approximately 8% in models with controls and matching. This effect underscores the critical role healthcare access plays in mental health outcomes. Similarly, experiencing the loss of loved ones also has a significant effect on the probability of PSS, with ATEs ranging from 9% to 11% across different models. This result underlines the profound emotional and psychological toll of such losses. Interestingly, while the effect of having infrequent contact with people in social networks appears to be not

**Table 5. Marginal effects analysis of experiencing diminished quality of life factors under Taliban 2.0 on Afghan adults residing in-country, post US military withdrawal.**

| Variables | Unadjusted | Controls | Matching | C+M |
|---|---|---|---|---|
| Can't buy food | -0.02 | -0.02 | 0.01 | 0.00 |
| (never 0–2 a lot) | (0.03) | (0.04) | (0.05) | (0.05) |
| No healthcare | 0.08** | 0.08** | 0.07* | 0.07 |
| (never 0–2 a lot) | (0.04) | (0.04) | (0.05) | (0.05) |
| Threat of violence | 0.36*** | 0.35*** | 0.35*** | 0.34*** |
| (never 0–2 a lot) | (0.02) | (0.02) | (0.03) | (0.03) |
| Loss of loved ones | 0.09** | 0.10** | 0.10* | 0.11** |
| (never 0–2 a lot) | (0.04) | (0.04) | (0.05) | (0.05) |
| Infrequent contact | 0.05 | 0.05 | 0.06 | 0.06 |
| (never 0–2 a lot) | (0.03) | (0.04) | (0.04) | (0.04) |
| Observations | 590 | 590 | 390 | 390 |
| Log Likelihood | -303.93 | -302.28 | -201.81 | -199.28 |
| Akaike Inf. Crit. | 619.86 | 622.57 | 415.62 | 416.57 |

Note: Binary-coded factors. *p<0.1

**p<0.05

***p<0.01.

statistically significant, the ATEs suggest a slight increase in the probability of increased PSS. This finding points to the potential mental health implications of social isolation. Yet, the most prominent results appear to be in experiencing the threat of violence, which has a robust and highly significant effect across all models. The ATEs suggest that experiencing threats of violence leads to a substantial increase in the probability of PSS, ranging from 34% to 36%.

The utilization of Marginal Effects in our analysis offers a clearer and more direct interpretation of the factors influencing bi-PSS (self-reported stress and no stress). By transitioning from traditional regression coefficients to ATEs, we provide insights that are both statistically rigorous and readily accessible, paving the way for targeted interventions and policies to address the identified risk factors. This methodological advancement underscores our commitment to delivering actionable research findings with direct implications for improving mental health outcomes. Overall, the results suggest that identified effects hold up, even when adjusting for multiple alternative causal channels.

## Discussion

### Shifts in structural determinants

Worldwide, structural determinants of health regularly benefit affluent populations, including people in power, wealthy households, or mainstream gender, religious, or ethnic groups. However, conflict often can quickly undo standing SDOHs that once benefitted specific populations like elites. In a conflict, power dynamic shifts can further redistribute vulnerability of hardships (like poverty, food insecurity, and violence) to affect not only marginalized populations but people who have lost political control [58], which appears to be the case in Afghanistan after 2021 The wake of the American military withdrawal may have been so seismic that prior structural determinants like gender and age that benefited mostly young men to prosper compared to those with lower social status, like women, appear to have washed away. As the humanitarian conditions lingers, even Afghan men, including those who are pro-Taliban,

seem to be facing DQOL and stress comparable to lower social status populations. First we must understand the SDOH in Afghanistan before 2021.

A review of the literature on Afghanistan implies since the Russian Afghan conflict to 2021, there were some consistent, key structural mechanisms (mainly gender, age, and steady income) that positively influenced Afghan individuals' socio-economic circumstances. These structural determinants historically benefited mostly men and young adults, while limiting SDOH particularly for female citizens, including elderly women [7,31,32,38,59–63]. Under the first Taliban rule (1996–2001), diminished social status tended to relate to extreme religious and misogynistic policies enforcing gender apartheid and gender-based violence (GBV). Female citizens were restricted to the home, less likely to be educated, had little to no health-care access or legal rights compared to men. These inequities were compounded by age. Afghan women tend to have a higher probability of diminished mental illness than men due to stigmatization and less access to care [8,30,31,45,64,65]. Additionally, age-related physical health complications (i.e. high blood pressure) and additional physical hardships (like violence and food insecurity), which many experienced in previous wars often compound DQOL [22,34–36,66]. Elderly survivors of war frequently experience higher prevalence rates of cognitive impairment, comorbid disorders including head injuries and depression, and medical comorbidities [37]. Yet, as the Old Taliban fell, some SDOH shifted.

During the American war in Afghanistan (2001–2021), gender and age-related inequalities changed in many regions. Women were granted increased human rights, including access to education, healthcare services, improved social mobility, political rights and participation, and employment, through programming often funded by multilateral and bilateral agencies like UNHCR and USAID. Improved healthcare services have reached more diverse populations including older adults with age-related health conditions [2,8,22]. However, these changes were unstable without Western-backed support. Many SDOH equity gain in the last twenty years were dramatically reversed under the Taliban 2.0's Islamic Emirate of Afghanistan. Many gender apartheid policies are reinstated, political-economic instability is on the rise, and access to basic needs and public services are greatly constrained [1,7,8,13,15,16,19,21,67]. While the Taliban retook power, the current dire conditions in Afghanistan appear to challenge even those who once socially benefitted under extreme Sharia law. Few SDOH are left unscathed.

Initial reports after the American withdrawal estimated that the political chaos would compound socio-economic hardships. In early 2022, estimates of the population currently under the poverty line ranged from 90–97% [13,14,68] and over half the population needing immediate humanitarian support [14,68]. Within the power vacuum created by the withdrawal, civil conflict increased between the Taliban and ISIS factions, including factions of the Pakistani Taliban, known as Tehrik-i-Taliban Pakistan (TTP); the Pakistani militants of Lashkar e Taiba; and the Islamic Movement of Uzbekistan. Arising ISIS cells in Afghanistan tend to be mainly made up of former Pakistani Taliban members [69,70].

Our 2023 data analysis indicates that men and young adults (18–35) are experiencing high levels of DQOL at an unexpected level. Previous studies tend to focus on traditionally more vulnerable populations like refugees, women, elderly, and children. Our literature review yielded few studies that assess the QOL shifts of men, most of whom tend to have better social status, access to education, political participation opportunities, social mobility, income generation, and serve as the decisionmaker over those within his household [9,16,36,59,71]. Yet, Afghan men in this study self-report experiencing the same levels of DQOL and PSS as women and older populations. The sample of female respondents and older peoples was limited; thus, we are cautious in drawing conclusions due to sampling biases. Yet, most of the qualitative feedback provided by the 873 participants further triangulates the initial finding that the

evolving crisis environment since 2021 may have shifted so much that the positive effects of these pre-US withdrawal structural mechanisms may have been washed out.

> This [new] stage has a deep and negative impact on people's lives. . . We [Afghans] don't live, we keep ourselves alive. [translated from Dari]– 20-year-old man

> The biggest challenge is unemployment, lack of proper food. . .gender contradictions, closing access to education for women, we face a new challenge every day [translated from Dari]– 32-year-old woman

> We think that the humanitarian crisis will increase, and the Afghans will face more problems [translated from Pashto]– 41-year-old man

> This is such a clarity that we all live a very hard life, there is no way to clear away our problems and fulfil our basic needs [translated from Pashto]– 21-year-old man

The bleak conditions in the country have worsened in such a way that nearly everyone irrespective of their background appears to be negatively affected.

## Behavioral & health system determinants

Structural determinants informing *Intermediary Determinants of Health* can crucially determine what kinds of vulnerability that individuals face, and moreover, to what extent they experience negative factors [24]. The data analysis indicates that nearly all Afghan adults are experiencing high rates of physical hardships related to IDH behavioral & health system factors, with food insecurity, limited healthcare access, and threat of violence being the most pressing.

Recent scholarship and reports on the state of Afghanistan regularly focus on food insecurity affecting millions [4,5,14,20,72] and the dire state of the health system [4,5,7,13,20,21]. Abbasi (2022) estimates that approximately 10 percent of Afghan babies born in 2022 died, a significant increase due to the spread of malnutrition, disease, and the healthcare sector decline. That same year, it is estimated that 95 percent of the population had irregular access to food, and millions of children needed nutritional support [11]. In our study, both statistical analysis and qualitative statements by many respondents indicate that food insecurity continues in 2023. 72.9% (CI95% 69.8–76.0) of respondents self-report having limited to no food access. They further paint a bleak picture of food shortages, including child-related hunger and deaths.

> There is no humanitarian aid coming. Our own people have not received even a kilo of flour. The Taliban have divided the nation. . . [translated from Pashto]– 30-year-old woman

> We cannot provide adequate food and clothing for our families [translated from Pashto]– 44-year-old man

> Lack of enough food causes malnutrition, especially in mothers and children [translated from Pashto] -27-year-old man

> Every day we see thing[s] that you never wish to see. Children dying, suffering no food, no money, no freedom [written in English]– 34-year-old man

The multivariate regression analysis indicates that food insecurity significantly affects PSS levels but depending on the modelling. The marginal effects analysis finds that food insecurity

in general may not play a significant factor on whether a person experiences psychosocial stress.

Comparatively, 71.6% (CI 95% 68.3–74.8) report experiencing little to no access to needed healthcare, including clinics. In regions where clinics or hospitals are still open, there appears to be insufficient medical supplies and healthcare professionals.

> I was a teacher, and I lost my job due to the arrival of the Taliban. . .My father was sick, and I was told that treatment is impossible here. I lost my father on the way [to another hospital] [translated from Pashto]– 21-year-old man

> Since Taliban took over all Afghanistan, I got jobless till now. . . My wife got sick. [S]he has tumor, and she needs an operation[,] and I am unable to help her due to financial problems. And I am losing her day by day. Live [living] is getting hard. And as we worked with U.S. government, and we are still in Afghanistan. . .Living here is like living in hell [in English]– 20-year-old man

The multivariate and matching analyses both confirm that the lack of healthcare access appears a strong indicator of experiencing PSS, including higher levels of stress. Particularly among male Afghan adults, there is a significant link between the extent to which he can access health provisions for his family and experiencing higher PSS levels ($\chi2 = 117.10$, $p<0.001$). A thorough analysis of the qualitative statements triangulates this finding, as the great majority of men tie concerns about lack of healthcare access and food insecurity to their inability to find stable employment. Statements by men tend to express a sense of shame at not fulfilling his duties as a provider, frustration at watching his family face poverty and hunger, and chronic anxiety about finding any means to take care of those who they love, all to the point of distress.

> From the moment Taliban took over all Afghanistan. All the situations has changed. The poverty level has grown up and most of the people are starving, they don't have job or work to do, they can't help their families financially. . . I have been unemployed since Taliban took over all Afghanistan. I can't find a job. They don't accept those people who worked with foreigners. Specially when me and my dad we both worked with U.S government in Afghanistan. The li[f]e is getting harder to live in it [English]– 21-year-old man

## Material determinants

There is a clear association between the loss of behavioral determinants, health system determinants, and material determinants (mainly the ability to work, be allowed to work, and to financially provide). Even men who have some stable employment, mainly those working for the Taliban government, express statements filled with sentiments of suffering and powerlessness in the inability to adequately provide and safeguard their household.

> In general, the security [under the Taliban Islamic Emirate of Afghanistan] is a bit better, but many educated young people are unemployed and looking for work, but unfortunately, they cannot find work, and Dubai labor dollars are smuggled to foreign countries, which face many problems [translated from Pashto] - 46-year-old New Taliban government worker, male

> The lack of unemployment is the psychological stress on my brain [translated from Pashto]– 44-year-old man, pro-Taliban views

Adults often suffer emotional and physical pain caused by disruptions in facing chronic poverty and violence, particularly in conflicts. Diminished social life conditions causes a sense of hurting, which becomes part of one's social experience [10,73–75]. "[P]overty and powerlessness are just as salient as war events in shaping experiences of trauma" [74] In Afghanistan, *ghairat/akhlaq* (morals governing appropriate behavior) is culturally valued, in which doing what is expected, working hard, following social norms, and respecting traditions are all tied to upholding their family honor and respect of their community. "Good *ghairat*" is an indicator of worthiness and character. To lose *ghairat* often is linked to economic hardship and domestic conflict [73]. Our study indicates that the economic hardships are becoming a shared source of trauma particularly for Afghan men, which in the long-run may lead to escalations in domestic violence, mental health distress, and poor efficacy.

An additional behavioral determinant of health that is mentioned in many qualitative statements by male and female respondents is the issues with gender apartheid restricting their physical mobility. The social constraints placed on women are intrinsically correlated to material IDH and psychosocial IDH, in which social discrimination limits workspace opportunities and social support networks.

> Women mostly being treated like pets. And they get all the pressure and not [allowed] going out from home. They are forced to wear "hijab" – 20-year-old man

> Women are [held] prisoner in Afghanistan. They can't go out, they can't work outdoor. They can't leave their house alone. They can't open hair salon shops. They can't fight back for their freedom, they are being raped and tortured. They don't count women [a]s a human [in English]– 21-year-old man

Women who are single, widowed, and/or the primary caregiver also face the traumatic experiences of wanting to fulfil the expectations to care for family, and the failure of doing so tied to sense of a loss of *ghairat* or a feeling of *be-ghairati*. Under the Americans, women seeking education and employment became more normalized; while under the Taliban, it has always been restricted, garnering a reemergent clash in gender norms and morality [71–73,76].

> I am taking care of my three siblings. My twelve-year-old brother does little work to get daily food, and I weave carpets with my brother and sister. On the one hand, there is an economic problem, and on the other hand, Talban does not allow girls [to work now] . . . everyone's life is a prison for me [translated from Dari]– 20-year-old woman

Nearly every respondent who provided qualitative statements remarks on the dire economic conditions tied to unemployment, with no prompts within the survey questions. There is a marked gap between Afghans who favored the daily life under the American-led forces, compared to those who favor the rule of law under the Taliban. Pro-American citizens frequently link their inability to find stable employment to social-economic discrimination that they face by the new government, concerns about corruption, and an increased worry of violent retaliation.

> [T]he current situation in Afghanistan is disastrous and economically unusual because there are no jobs, all government jobs are taken by the Taliban, and the private sector has no absorption capacity. The Taliban are armed in the city. They do not know whether they are thieves or responsible [translated in Pashto] - 40-year-old man

On the other spectrum, many Afghans, including some women, do not fault the New Taliban government. Instead, they often remark on how Afghanistan feels more 'secure' and 'peaceful' due to the 2021 transition.

> Currently, there is better security in Afghanistan, but the economic situation of the people is weaker than before [translated from Pashto] -21 year-old male student

Despite whatever their political views, an initial qualitative assessment of all Afghan open-ended responses indicate that many (pro-Taliban/pro-Republic) feel that the humanitarian crisis is affecting everyone. They often place blame on the sanctions.

> Some international sanctions have created economic difficulties [translated from Dari]- 21-year-old woman

> If the international community removes the economic sanctions from Afghanistan and recognizes the government at the international level, Afghans can live a better life [translated from Pashto]– 30-year-old man

> In general, the Taliban people are supportive, but despite the difficulties, the effect of the weak economy has come. The United Nations and the United States are all to blame [English]– 30-year-old man

Following the Taliban takeover, Afghanistan continues to face an international freeze on most developmental and governance aid and assistance. Additionally, a freeze on the Afghan State Bank and related assets based in the US and Europe have weakened both private and public sector and led to large-scale unemployment. Moreover, commodity prices have surged. The Taliban- Islamic Emirate of Afghanistan's 2022 renewal of gender apartheid policies has further depleted the workforce and economic participation, as well as served as the central catalyst for remaining INGOs, bilateral and multilateral stakeholders to pull programming and funding support [5,11,14,15,67].

Yet, there are concerns that tightening restriction on the new government by the international community will not work against the Taliban, but instead further drive dire humanitarian conditions in country [77–80]. New sanctions should be cautioned. Lifting existing sanctions could help improve conditions on the ground, preventing needless starvation, deaths, and a thrust towards more extremist isolationism within the country. Navigating the nuanced boundary wherein assistance effectively reaches populations entrenched in chronic poverty, without inadvertently sustaining oppressive regimes that impede economic progress and self-reliance through extremist policies, constitutes a formidable challenge necessitating meticulous attention. Since 1970, sanctions proved effective approximately 13 percent of the time. "In most cases, sanctions have not only failed to achieve their stated goals but have also backfired, harming U.S. interests and emboldening the sanctioned entity" [80].

## Psychosocial circumstances

Our statistical analysis highlights the severe impact that the threat of violence and bodily harm can have on psychosocial social stress & mental health (MHPSS). Approximately, six-in-ten Afghan adults face some threat of harm or violence, of which is most highly correlated with experiencing extreme sadness, anxiety, and nightmares (r>.50). While many agree that daily life has become more stable since the 2021 takeover, there are ongoing concerns of external violence. An analysis of the testimonies from the 873 respondents helps inform the causal

model behind the threat of violence and experiencing higher levels of stress. Two main themes emerge in relation to the source of the threat of violence and physical harm.

Firstly, many Afghans respondents indicate that the American withdrawal has led to a new kind of domestic security in everyday life. Internal conflict has diminished.

> We are glad of currently government behaviors. . .With coming of them [the Taliban-Islamic Emirate of Afghanistan]: No War, No stealing, No kidnapping– 24-year-old man

> When we went to the city for shopping or sightseeing [before 2021 withdrawal], there was a fear that a bomb would explode here or a suicide attack would take place here, so this fear is not present now, it is safe, thankfully because of the Taliban winning– 23-year-old man

They tend to give credit to the Taliban government for stopping suicide bombings and other forms of terrors instilled throughout the country and major cities before their takeover. They also give credit to the Taliban government as trying its best to run the country despite circumstances. But these testimonies may be affected by some bias, such as some participants wanting to provide a more positive view of the new government, framing the Taliban as heroes due to their political leanings. Yet even pro-Taliban respondents indicate the severity by which failure to follow new religious laws will be punished.

> The Taliban are drowning in [Afghan people's] love. They are now doing favors [for] the nation. We [us supporters of the Taliban] brought the Islamic system. We did it. We did it. We will do whatever we do. But you [who question the Taliban] must apply the full Islamic laws. Otherwise, they will hang us. [T]hey say we have defeated the world, and everything should be in our hands, and we will use it for love and four marriages. You are apostates, be patient. My heart is full [translated from Pashto]– 35-year-old man

In this particular statement, the respondent differentiates 'they' as the Taliban, 'we' as their supporters, and 'you' as apostates or non-believers. The contradictions of both being proud of the new Islamic changes but the threat of death of not only apostates but 'us' is concerning. Several others mention fear of violence from external government sources related to Pakistani-Taliban/ISIS cells. Those that mention facing bombs allude that they are not detonated by the New Taliban forces, but instead ISIS factions. Furthermore, they seem to perceive that the economic hardships, humanitarian crisis, and lack of international support for the New Taliban regime may eventually result in increased destabilization, leading to more extremist zealotism.

In contrast, for each pro-Taliban statement, there are an estimated two respondents who express fear of the Taliban- Islamic Emirate of Afghanistan, its workers, and fighting force. Most of these people mention working for the Western allies including the US military, Western INGOs, UN agencies, or President Ashraf Ghani's government. Others hold Western-based education, support gender equity, or did not actively support the Taliban.

> [W]e are facing many problems, we do not have freedom of speech, economic problems are too much, girls are not allowed to study and those who worked with the previous government are tortured and many of them were killed by the Taliban [English]– 22 year-old man

> [There is] increased pressure on civilians. [T]hey [are] being prosecuted, people in my country are being treated partially for being involved in the previous government [English]– 24-year-old man

A 2023 UN report indicates that people affiliated with the former government and security forces of Afghanistan are increasingly targeted by Taliban agents, despite initial promises that no reprisals would be carried out. In contrast, there are reports of mostly men being executed without trial, arbitrarily arrested, detained, and/or torture since the Taliban takeover. These issues do not pertain to most of the Afghan population. But most Afghans are believed to be facing increased controls, intimidation, and threat of violence [18,19]. The qualitative feedback from respondents indicates that while democratic freedoms and economic prosperity have decreased, security in cities and villages has increased, at least for the Taliban and their supporters. The threat of violent retaliation or reprisals by Taliban fighters against those who supported the Republic under the Americans is tangible. Yet, the stability established under the Taliban 2.0 is also at threat by external and internal conflict involving ISIS cells. But, in general, this study's findings indicate that the threat of bodily harm or violence may not be as commonplace as issues like food insecurity or healthcare access, but statistically it leads to a much higher likelihood of experiencing PSS. Secondly, the more frequently that a person experiences threat of harm, the more extensive their stress levels are.

## Limitations

The snowball survey approach of this study limits generalizability. We try to address this deficient through extensive statistical analysis including matching. Given the increased research difficulties within the country, digital data collection was our only alternative. Additional limitations include the small sampling of subpopulations of female and older respondents, which limited comparability. Our study also purposefully selected to not include additional demographics like ethnicity, language, and location in the formal analysis for ethical concerns. In concerns to our matching technique, while the inclusion of additional variables could enhance the depth of analysis, the constraints of the current environment necessitated a cautious approach. Our non-ideal methodological choices were mainly influenced by several factors.

Firstly, we consider the ethical and safety considerations in data collection. In the current Afghan context, certain questions that might be benign elsewhere can pose significant risks to respondents. As presented, our research team determined not to inquire about specific locations, languages, or ethnicities. Pilot-testing feedback indicated that this could lead to discomfort or fear due to historical conflicts and the prevailing political climate. Academic literature supports this caution. For instance, Blair, Imai, and Lyall (2014) discuss the challenges of collecting sensitive data in conflict zones, emphasizing the need to protect respondent anonymity to ensure safety. Their other work (2013) highlights the importance of survey experiments to control biases and the risks such variables may introduce in observational data, as in our case. Using indirect questioning techniques can mitigate risks in sensitive environments. Additionally, we consider optimizing the questionnaire length for enhancing response rates. We aimed to balance the depth of information with respondent engagement by limiting the questionnaire length. Research indicates that in environments like Afghanistan under the Taliban, shorter surveys tend to achieve higher response rates. For example, Galesic and Bosnjak (2009), Fan and Yan (2010), and Holtom et al. (2022) found that longer questionnaires significantly decrease participation rates in similar contexts. Adjusting for additional regional-level and individual-level factors would improve the transparency of our results and might provide further insights into the mechanisms we study in this paper. As noted, future studies may consider including additional demographic variables. Including an "education" variable could have offered especially valuable insights for our study, and we regret not incorporating it into the questionnaire perhaps at the price of a lower response rate. Lastly, our paper utilizes the qualitative data for "triangulation," with qualitative findings secondary to quantitative

outcomes. This paper seeks to first establish statistical trends, while mainly using quotes to illustrate quantitative trends. Future research will provide a more in-depth analysis of the qualitative data, using a narrative approach.

## Research implications

This study is one of the first studies to extract data from within Afghanistan since 2022 when the Taliban took full control over the government. The findings indicate that prior structural determinants like gender and age that benefited mostly young men to prosper compared to those with lower social status, like women, may have ceased in the wake of American withdrawal and the rise of the Taliban- Islamic Emirate of Afghanistan. These dynamic shifts in SDH at the primary level may be temporary. But in 2022–23, most Afghan men report facing one or more DQOL factor, related to economic hardships stunting their ability as the male provider to adequately provide for his family, resulting in a loss of *ghairat*. Psychosocial stress like anger and anxiety, which derive mainly from material determinants, are the outcome of what appears a shared trauma among men. Likewise, the new gender apartheid policies are linked to diminished quality of life for women and girls.

The snowball survey approach of this study limits generalizability. We try to address this deficiency through an extensive statistical analysis including matching. Given the increased research difficulties within the country, digital data collection was our only alternative. Additional limitations include the small sampling of subpopulations of female and older respondents, which limited comparability. Our study also purposefully selected to not include additional demographics like ethnicity, language, and location in the formal analysis for ethical concerns. Future research may address some of these limitations. Yet, our research offers unique quantitative statistics measuring the extent of DQOL and PSS factors, as well as extensive qualitative data that triangulates the statistical data and helps inform the causal relationships between the variables, based on the experiences of those most intimately affected by the humanitarian crisis.

## Conclusions

This research validates many of the concerns of the humanitarian crisis on the ground, as well as provides insight into how political shifts have resulted in socio-economic hardships affecting Afghans who remained in country after the 2021 US withdrawal. Our study indicates that most Afghans living under the Taliban 2.0, regardless of their political allegiances or demographics, are suffering in some way, leading to high levels of psychosocial stress. We offer policy recommendations based on the literature review, quantitative and qualitative results of our survey. Firstly, the issues of food insecurity and insufficient healthcare access are affecting many, resulting in needless suffering, poverty, and deaths including small children. The loss of economic activity, propelled in many ways by international sanctions and restrictions, may be ineffective against the Taliban, but instead negatively affects the everyday Afghan citizen. The lack of active employment is linked to higher rates of physical and emotional hardship, especially among men, which may lead to further health complications, domestic violence, and extremism over the long term. The data analysis reveals dire conditions facing most Afghan respondents, mostly educated men. Conditions of DQOL and PSS are likely even more desperate for Afghans not reached by this study, such as children, women with limited education, and rural households with limited ICT.

It is advisable for the US and European allies to consider eliminating current sanctions, and working with international humanitarian actors like UNICEF, IRCR, and the WHO to provide supplemental aid and assistance. This decision may require Western actors to conciliate with

some Taliban policies that they do not normally support, such as operating within some parameters of gender apartheid. Secondly, there is an urgent need for the Taliban government, state actors, and international stakeholders to address the MHPSS needs of its citizens. Public health solutions should provide culturally appropriate and contextually relevant programming. Thirdly, more research of in-country populations is needed to help inform our understanding of conditions affecting citizens as the conflict evolves, and to help inform meaningful policy and programming solutions. Lastly, as the years have passed, and new conflicts like the Ukraine-Russian and Israeli-Hamas wars emerge, the humanitarian crisis in Afghanistan has received decreasing global attention. As an international community, we should strive to not forget those who are suffering under dreadful humanitarian conditions and ongoing threat of violence.

## Supporting information

**S1 Table. Pairwise correlation analysis of 2022–23 Afghan psychosocial stress and quality-of-life.**
(DOCX)

**S1 Fig. Discovery analysis.**
(TIF)

**S1 Text. Afghan survey questions.**
(DOCX)

## Acknowledgments

Thank you to Dr. Cat Worsnap and Dr. Donal Skinner for your support and feedback. This research is dedicated to the people of Afghanistan, their struggle, their love, and their hope for their nation.

## Author Contributions

**Conceptualization:** Jessi Hanson-DeFusco, Alexis McMaster, Hamid Popalzai, Heer Shah, Min Shi, Nandita Kumar.

**Data curation:** Jessi Hanson-DeFusco, Anton Sobolov, Sami Stanekzai, Hamid Popalzai.

**Formal analysis:** Jessi Hanson-DeFusco, Anton Sobolov.

**Investigation:** Jessi Hanson-DeFusco, Sami Stanekzai, Hamid Popalzai, Heer Shah, Min Shi, Nandita Kumar.

**Methodology:** Jessi Hanson-DeFusco, Anton Sobolov, Alexis McMaster, Hamid Popalzai, Heer Shah, Min Shi, Nandita Kumar.

**Project administration:** Jessi Hanson-DeFusco, Sami Stanekzai, Hamid Popalzai, Heer Shah, Min Shi, Nandita Kumar.

**Software:** Jessi Hanson-DeFusco, Anton Sobolov.

**Supervision:** Jessi Hanson-DeFusco.

**Validation:** Anton Sobolov, Sami Stanekzai, Alexis McMaster.

**Visualization:** Jessi Hanson-DeFusco, Anton Sobolov, Nandita Kumar.

**Writing – original draft:** Jessi Hanson-DeFusco, Anton Sobolov, Alexis McMaster, Heer Shah.

**Writing – review & editing:** Jessi Hanson-DeFusco, Sami Stanekzai, Alexis McMaster, Hamid Popalzai, Heer Shah, Min Shi, Nandita Kumar.

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
