## [Decision Letter · Decision Letter 0]

5 Nov 2024

PMEN-D-24-00296

The association of diminished quality of life of Afghan adults’ psychosocial wellbeing, in the era of the New Taliban Government

PLOS Mental Health

Dear Dr. Hanson-DeFusco,

Thank you for submitting your manuscript to PLOS Mental Health and thank you for your patience whilst we reach a decision. After careful consideration of the reviewer comments, we feel that your paper has merit but does not fully meet PLOS Mental Health’s publication criteria as it currently stands. Therefore, we invite you to submit a revised version of the manuscript that addresses the points raised during the review process.

Please address all of the reviewer comments, which you can find at the end of this email.

We look forward to receiving your revised manuscript.

Kind regards,

Karli Montague-Cardoso

Executive Editor

PLOS Mental Health

Journal Requirements:

1. Please include a complete copy of PLOS’ questionnaire on inclusivity in global research in your revised manuscript. Our policy for research in this area aims to improve transparency in the reporting of research performed outside of researchers’ own country or community. The policy applies to researchers who have travelled to a different country to conduct research, research with Indigenous populations or their lands, and research on cultural artefacts. The questionnaire can also be requested at the journal’s discretion for any other submissions, even if these conditions are not met.  Please find more information on the policy and a link to download a blank copy of the questionnaire here: https://journals.plos.org/plosmentalhealth/s/best-practices-in-research-reporting. Please upload a completed version of your questionnaire as Supporting Information when you resubmit your manuscript. 2. Please amend your detailed Financial Disclosure statement. This is published with the article. It must therefore be completed in full sentences and contain the exact wording you wish to be published. **Please only choose the relevant sentences from below** 1. Please clarify all sources of funding (financial or material support) for your study. List the grants (with grant number) or organizations (with url) that supported your study, including funding received from your institution. 2. State the initials, alongside each funding source, of each author to receive each grant.3. State what role the funders took in the study. If the funders had no role in your study, please state: “The funders had no role in study design, data collection and analysis, decision to publish, or preparation of the manuscript.”4. If any authors received a salary from any of your funders, please state which authors and which funders. If you did not receive any funding for this study, please simply state: “The authors received no specific funding for this work.” 3. We note that you have indicated that there are restrictions to data sharing for this study. For studies involving human research participant data or other sensitive data, we encourage authors to share de-identified or anonymized data. However, when data cannot be publicly shared for ethical reasons, we allow authors to make their data sets available upon request. For information on unacceptable data access restrictions, please see http://journals.plos.org/plosone/s/data-availability#loc-unacceptable-data-access-restrictions.   Before we proceed with your manuscript, please address the following prompts:  a) If there are ethical or legal restrictions on sharing a de-identified data set, please explain them in detail (e.g., data contain potentially identifying or sensitive patient information, data are owned by a third-party organization, etc.) and who has imposed them (e.g., a Research Ethics Committee or Institutional Review Board, etc.). Please also provide contact information for a data access committee, ethics committee, or other institutional body to which data requests may be sent.  b) If there are no restrictions, please upload the minimal anonymized data set necessary to replicate your study findings to a stable, public repository and provide us with the relevant URLs, DOIs, or accession numbers. Please see http://www.bmj.com/content/340/bmj.c181.long for guidelines on how to de-identify and prepare clinical data for publication. For a list of recommended repositories, please see https://journals.plos.org/plosone/s/recommended-repositories. You also have the option of uploading the data as Supporting Information files, but we would recommend depositing data directly to a data repository if possible.  Please update your Data Availability statement in the submission form accordingly. 4. Please provide separate figure files in .tif or .eps format. For more information about figure files please see our guidelines:   https://journals.plos.org/mentalhealth/s/figures https://journals.plos.org/mentalhealth/s/figures#loc-file-requirements  5. Tables should not be uploaded as individual files. Please remove these files and include the Tables in your manuscript file as editable, cell-based objects. For more information about how to format tables, see our guidelines:  https://journals.plos.org/mentalhealth/s/tables 6. We have noticed that you have uploaded Supporting Information files, but you have not included a list of legends. Please add a full list of legends for your Supporting Information files after the references list.  7. We notice that your supplementary figures are uploaded with the file type 'Figure'. Please amend the file type to 'Supporting Information'. Please ensure that each Supporting Information file has a legend listed in the manuscript after the references list. 

Additional Editor Comments (if provided):

Reviewers' comments:

Reviewer's Responses to Questions

**Comments to the Author**

1. Does this manuscript meet PLOS Mental Health’s publication criteria? Is the manuscript technically sound, and do the data support the conclusions? The manuscript must describe methodologically and ethically rigorous research with conclusions that are appropriately drawn based on the data presented.

Reviewer #1: Yes

Reviewer #2: Yes

2. Has the statistical analysis been performed appropriately and rigorously?

Reviewer #1: I don't know

Reviewer #2: Yes

3. Have the authors made all data underlying the findings in their manuscript fully available (please refer to the Data Availability Statement at the start of the manuscript PDF file)?

Reviewer #1: Yes

Reviewer #2: Yes

4. Is the manuscript presented in an intelligible fashion and written in standard English?

Reviewer #1: No

Reviewer #2: No

5. Review Comments to the Author

Reviewer #1: Thanks for the authors for undertaking such an important subject, here are some more pints for its clarity and fluency.

Keywords: The term "new Taliban" may be misleading; it might be more appropriate to refer to this as the "new Taliban era." While I understand the intention behind "new," this phrasing could lead to confusion.

Abstract:

- Results: The last sentence regarding qualitative data would be better placed in the Methods section rather than in Results. Additionally, the conclusion is quite general; it would benefit from being more specific to the findings of this study.

Introduction:

- NATO should be spelled out upon first use.

- There are several sentences that begin with numbers in both the abstract and the body. For improved readability, I suggest writing out the numbers or using a word before them.

- The phrase "the purpose of this study" would fit better as a subsection of the Methods section. It’s helpful to have a general statement of purpose at the end of the introduction, but detailed information is best kept in the Methods section.

Methods:

- In the participants section (page 7), some results are mentioned. It would be more appropriate to focus solely on the methods and avoid including results in this part.

The best

Reviewer #2: Dear authors,

This is an important contribution to the literature given the paucity of data on Afghanistan. I have a few comments that I would like your clarification on.

1: I'm curious about the absence of a reflexivity statement. Given the sensitive sociopolitical context of Afghan adults' quality of life under the Taliban regime, including such a statement would provide transparency regarding the authors' relationship to the context. An awareness of researchers' positionality can better inform readers about potential influences on data collection and analysis.

2: Could you provide more information about the questionnaire development process? Specifically:

a) What sources were used to develop the questionnaire?

b) How was it piloted?

c) Were internal and external validity assessed?

d) Did the scales undergo cultural adaptation?

3: Regarding the qualitative surveys:

a) How were they conducted?

b) Who conducted them?

c) Are the interviewers' contributions acknowledged?

d) What tools were used for qualitative data analysis?

4: While you employed matching techniques to control for confounding variables, a more detailed justification for the selection of specific control variables is essential. For instance, controlling for region (urban vs. rural), education level, or political affiliations could add depth, as these factors might directly impact stress levels under the current political regime.

5: The results mention qualitative data "triangulation," but the integration appears limited, with qualitative findings seeming secondary to quantitative outcomes. Please consider expanding the qualitative results section or using quotes to illustrate quantitative trends. This would add richness and narrative depth, particularly given the study's mixed-methods nature.

Good luck!!

6. PLOS authors have the option to publish the peer review history of their article (what does this mean?). If published, this will include your full peer review and any attached files.

**Do you want your identity to be public for this peer review?** For information about this choice, including consent withdrawal, please see our Privacy Policy.

Reviewer #1: No

Reviewer #2: No

---

## [Editor Report · Decision Letter 1]

25 Nov 2024

The association of diminished quality of life of Afghan adults’ psychosocial wellbeing, in the era of the Taliban 2.0 Government

PMEN-D-24-00296R1

Dear Dr. Hanson-DeFusco,

We are pleased to inform you that your manuscript 'The association of diminished quality of life of Afghan adults’ psychosocial wellbeing, in the era of the Taliban 2.0 Government' has been provisionally accepted for publication in PLOS Mental Health.

Best regards,

Karli Montague-Cardoso

Staff Editor

PLOS Mental Health